# A Photon Force and Flow for Dissipative Structuring: Application to Pigments, Plants and Ecosystems

**DOI:** 10.3390/e24010076

**Published:** 2022-01-01

**Authors:** Karo Michaelian, Ramón Eduardo Cano Mateo

**Affiliations:** 1Department of Nuclear Physics and Application of Radiation, Instituto de Física, Universidad Nacional Autónoma de México, Cto. de la Investigación Científica, Cuidad Universitaria, Mexico City C.P. 04510, Mexico; 2Facultad de Ciencias, Universidad Nacional Autónoma de México, Cto. de la Investigación Científica, Cuidad Universitaria, Mexico City C.P. 04510, Mexico; kner.409@ciencias.unam.mx

**Keywords:** dissipative structuring, non-equilibrium thermodynamics, entropy production, origin of life, organic pigments, plants, ecosystems, evolution, chlorophyll, biosignatures, 92-10, 92C05, 92C15, 92C40, 92C45, 80Axx, 82Cxx

## Abstract

Through a modern derivation of Planck’s formula for the entropy of an arbitrary beam of photons, we derive a general expression for entropy production due to the irreversible process of the absorption of an arbitrary incident photon spectrum in material and its dissipation into an infrared-shifted grey-body emitted spectrum, with the rest being reflected or transmitted. Employing the framework of Classical Irreversible Thermodynamic theory, we define the generalized thermodynamic flow as the flow of photons from the incident beam into the material and the generalized thermodynamic force is, then, the entropy production divided by the photon flow, which is the entropy production per unit photon at a given wavelength. We compare the entropy production of different inorganic and organic materials (water, desert, leaves and forests) under sunlight and show that organic materials are the greater entropy-producing materials. Intriguingly, plant and phytoplankton pigments (including chlorophyll) reach peak absorption exactly where entropy production through photon dissipation is maximal for our solar spectrum 430<λ<550 nm, while photosynthetic efficiency is maximal between 600 and 700 nm. These results suggest that the evolution of pigments, plants and ecosystems has been towards optimizing entropy production, rather than photosynthesis. We propose using the wavelength dependence of global entropy production as a biosignature for discovering life on planets of other stars.

## 1. Introduction

The dissipative process in which a part of the incident solar spectrum is absorbed and irreversibly converted into heat by organic or inorganic material at the Earth’s terrestrial surface, or in Earth’s atmosphere or oceans, is by far the most important entropy-producing process occurring on Earth. However, a generalized photon force for this irreversible process has not yet been given in the literature. Here, after deriving Planck’s formula for the entropy of an arbitrary beam of photons and from within the framework of the Classical Irreversible Thermodynamic (CIT) theory of the school of Ilya Prigogine [1,2,3,4], we derive the entropy production for this irreversible process as the product of a generalized force and flow, corresponding to a generalized thermodynamic force involving wavelength-dependent absorption and emissivity and the flow of incident photons towards the infrared spectrum.

CIT theory assumes a “local equilibrium” where the usual thermodynamic variables, as well as the Gibb’s relation between them, remain valid locally (in space and time) even though the global system may be far from equilibrium. A discussion of this assumption, in relation to the photon dissipative process, is given in an accompanying article in this Special Issue [5]. Other relevant frameworks for treating non-equilibrium systems are discussed elsewhere [6].

The photon force has utility in analyzing the dissipative structuring of pigments in plants, or the succession of ecosystems towards a climax and this has relevance to the *Thermodynamic Dissipation Theory for the Origin and Evolution of Life*, where the fundamental organic molecules of life are considered as dissipatively structured pigments obtained from simpler and more common precursor molecules (such as HCN in water) under the long-wavelength UVC surface spectrum of the Archean [5,7,8,9].

In this paper, we perform a wavelength-dependent grey-body calculation of the entropy production of different material (water, a sand and rock desert, a leaf and a climax forest) under sunlight. Our analysis suggests that material under the incident spectrum of our Sun will tend to self-organize, subject to restrictions arising from the physical properties of the material, in such a manner so as to become ever more like a black-body, i.e., approaching maximal short-wavelength photon absorption and maximal red-shifted long-wavelength emissivity.

We find that plant leaves have large entropy-producing potential per unit volume and that climax forests provide greater entropy production than deep ocean water (similar to black bodies). Finally, we suggest that plant and phytoplankton pigments (including chlorophyll) absorb maximally between 430 and 550 nm because this is exactly where entropy production due to photon dissipation is maximal for our solar spectrum, while photosynthetic efficiency is maximal between 600 and 700 nm. Therefore, we propose that, contrary to common belief, pigments, plants and ecosystems tend to optimize entropy production rather than photosynthesis.

Finally, we propose that the wavelength dependence of global entropy production can be used as a biosignature for discovering life on planets of other stars.

## 2. Derivation of the Planck Equation for the Entropy of an Arbitrary Photon Spectrum

The thermodynamics of photon (electromagnetic field) interaction with material is still an active area of research [10,11] that began with the work of Maxwell, Planck and Einstein at the turn of the previous century. In this section, we present the derivation of an expression, in modern notation, for the entropy of a photon beam of an arbitrary spectrum, as first obtained by Max Planck in 1913 [12].

Assume we have electromagnetic energy *E* within a square cavity of size *L* with perfectly conducting walls. The electric field must, therefore, go to zero at the walls. First, we will determine the number of resonant modes M(λ)dλ that can be supported inside the cavity for a given wavelength interval between λ and λ+dλ. Then, with this number and the number of ways that *P* energy packets each of energy ε=hν=hc/λ can be distributed over the number of available modes, M(λ)dλ, gives the number of microstates *W* for the partition of the energy and, therefore, from Boltzmann’s equation, we obtain the entropy as S=klnW. Later, in the derivation, we will remove the restriction of the confinement of the radiation to a box.

For standing (resonant) waves in a square box of dimension *L*, it is required that, $\frac{n}{2}\lambda =L$, where *n* is an integer value and λ is the wavelength of the resonant mode. Since there are three independent coordinates, we have three similar equations:(1)$$\left(n_{x},n_{y},n_{z}\right)/2\cdot \lambda =L=V^{1/3}.$$

The number of different combinations of the $n_{i}$ that satisfy these equations are the number of different standing wave modes which can be supported for wavelength λ inside the cavity of size *L* and volume *V*. To count this number, consider that it corresponds to the number of integral grid points internal to the positive octant of a sphere of radius *R* in an imaginary *n* space, such that
$$\left(n_{x}^{2}+n_{y}^{2}+n_{z}^{2}\right)\leq R^{2},$$
where, from Equation (Equation 1), $R=2L/\lambda =2V^{1/3}/\lambda$.

The number of lattice points inside this sphere, for a large sphere, is just the volume of this sphere $\frac{4\pi }{3}R^{3}$ (because grid or lattice points are equally spaced in all dimensions); however, the *n* can only be positive. Therefore, we take the volume of only the positive octant. There are also two polarization states available for unpolarized light, or one state for polarized light, i.e., $n_{0}=2$ or 1, respectively. Therefore, the total number of modes of wavelength λ is
(2)$$M\left(\lambda \right)=n_{0}\frac{1}{8}\frac{4\pi }{3}R^{3}=\frac{\pi n_{0}}{6}{\left(\frac{2V^{1/3}}{\lambda }\right)}^{3}=\frac{4\pi n_{0}}{3\lambda^{3}}V.$$

The number of modes dM within an interval dλ is then simply
(3)$$dM=\frac{dM}{d\lambda }d\lambda =\frac{-4\pi n_{0}}{\lambda^{4}}Vd\lambda .$$

We have, therefore, derived that we have M(λ) resonant modes in a volume *V* and we have a total energy E(λ) to distribute over these M(λ) modes. Energy is quantized and comes in packets ε=hν=hc/λ. Therefore, we have *P* packets of energy ε, such that E(λ)=P(λ)ε.

To calculate the entropy S(λ) of the light inside the volume, we use Boltzmann’s equation,
S(λ)=klnW(λ).

Note that for non-interacting photons (true at least at normal photon densities and temperatures) the probabilities of all microstates are the same and the Boltzmann and Gibbs definitions of entropy are identical.

We, therefore, need to calculate the number of microstates W(λ) which corresponds to the number of ways to distribute P(λ)=E(λ)/ε energy elements over M(λ) modes for carrying this energy. The number of ways to distribute *P* energy elements over *M* resonant modes is obtained using combinatorial theory:(4)$$W=\frac{(M+P-1)!}{(M-1)!P!}.$$

Since, for macroscopic volumes and not-too-long wavelength light, the number of allowed modes *M* and the number of energy packets *P* is very large, we can ignore −1 in the numerator and denominator. For the same reason, we can also use Stirling’s approximation,
ln(M!)=Mln(M)−M+O(lnM),
and, therefore, for the number of modes *M* large,
(5)$$M!≃M^{M}.$$

Using this approximation in Equation (Equation 4) gives the number of microstates as,
$$W=\frac{{\left(M+P\right)}^{M+P}}{M^{M}P^{P}},$$
and, therefore, using Boltzmann´s equation, the entropy of the photons in the box is
S(λ)=klnW(λ)=k[(M(λ)+P(λ))ln(M(λ)+P(λ))−M(λ)lnM(λ)−P(λ)lnP(λ)].

Since P(λ)=E(λ)/ε=M(λ)U(λ)/ε, where U(λ)=E(λ)/M(λ) is the average energy of the modes of wavelength λ, the entropy can be written as: (6)$$\begin{array}{ccc} S & = & k\left[\left(M+\frac{MU}{\varepsilon }\right)\ln\left(M+\frac{MU}{\varepsilon }\right)-M\ln M-\frac{MU}{\varepsilon }\ln\frac{MU}{\varepsilon }\right]\\ & = & kM\left[\left(1+\frac{U}{\varepsilon }\right)\ln M\left(1+\frac{U}{\varepsilon }\right)-\ln M-\frac{U}{\varepsilon }\ln\frac{MU}{\varepsilon }\right]\\ & = & kM\left[\left(1+\frac{U}{\varepsilon }\right)\ln\left(1+\frac{U}{\varepsilon }\right)-\frac{U}{\varepsilon }\ln\frac{U}{\varepsilon }\right], \end{array}$$
and, therefore,
(7)$$\frac{dS}{dM}=k\left[\left(1+\frac{U}{\varepsilon }\right)\ln\left(1+\frac{U}{\varepsilon }\right)-\frac{U}{\varepsilon }\ln\frac{U}{\varepsilon }\right],$$
where we have suppressed the dependence on λ for convenience.

Since E=MU, then dE/dM=U, and
(8)$$U=\frac{dE}{dM}=\frac{dE}{d\lambda }\frac{d\lambda }{dM}=\frac{-dE}{d\lambda }\frac{\lambda^{4}}{4\pi n_{0}V},$$
where we have used Equation (Equation 3) for deriving the last term.

Now, consider not the static situation of electromagnetic radiation in a box, but rather a beam of radiation moving at velocity *c* at intensity of energy flow i(λ) per unit wavelength interval dλ, per unit solid angle, per unit area of the source. In this case, we have,
(9)$$i\left(\lambda \right)=-\frac{dE}{d\lambda }\frac{1}{4\pi }\frac{c}{V},$$
where the factor of 1/4π comes in because i(λ) is to be given per unit solid angle and the factor of c/V comes in because we need to convert from energy per unit volume to energy flow per m${}^{-2}$ for a beam traveling at velocity *c*. For example, i(λ) has units (J m${}^{-3}$ s${}^{-1}$ sr${}^{-1}$) if λ and volume *V* are measured in meters. Using this in Equation (Equation 8) gives
(10)$$U\left(\lambda \right)=\frac{i\left(\lambda \right)\lambda^{4}}{n_{0}c}.$$

Similarly,
(11)$$\frac{dS}{dM}=\frac{dS}{d\lambda }\frac{d\lambda }{dM}=\frac{-dS}{d\lambda }\frac{\lambda^{4}}{4\pi n_{0}V},$$
where we have used, again, Equation (Equation 3) for the last term.

Since we are now considering not the entropy S(λ) of a given volume *V* containing electromagnetic radiation, but the flow of entropy j(λ) due to a flux of photons moving at the speed of light *c* with intensity i(λ) per unit wavelength interval, per unit time, per unit solid angle, per unit area,
(12)$$j\left(\lambda \right)=-\frac{dS}{d\lambda }\frac{1}{4\pi }\frac{c}{V},$$
where, as in Equation (Equation 9), the factor of 1/4π comes in because j(λ) is given per unit solid angle and the factor of c/V comes in because we need to convert from entropy per unit volume to entropy flow per m${}^{-2}$ for a beam traveling at velocity *c*. For example, j(λ) has units (J K${}^{-1}$ m${}^{-3}$ s${}^{-1}$ sr ${}^{-1}$) with λ given in meters.

Using Equation (Equation 12) in Equation (Equation 11) and Equation (Equation 10) together in Equation (Equation 7), with ε=hν=hc/λ, gives, finally,
(13)$$j\left(\lambda \right)=\frac{n_{0}kc}{\lambda^{4}}\left[\left(1+\frac{\lambda^{5}i\left(\lambda \right)}{n_{0}hc^{2}}\right)\ln\left(1+\frac{\lambda^{5}i\left(\lambda \right)}{n_{0}hc^{2}}\right)-\frac{\lambda^{5}i\left(\lambda \right)}{n_{0}hc^{2}}\ln\frac{\lambda^{5}i\left(\lambda \right)}{n_{0}hc^{2}}\right],$$
and the equivalent in terms of frequency ν is
(14)$$j\left(\nu \right)=\frac{n_{0}k\nu^{2}}{c^{2}}\left[\left(1+\frac{c^{2}i\left(\nu \right)}{n_{0}h\nu^{3}}\right)\ln\left(1+\frac{c^{2}i\left(\nu \right)}{n_{0}h\nu^{3}}\right)-\frac{c^{2}i\left(\nu \right)}{n_{0}h\nu^{3}}\ln\frac{c^{2}i\left(\nu \right)}{n_{0}h\nu^{3}}\right].$$

Note that nowhere in this derivation of the entropy of a beam of photons have we put restrictions on the type of radiation i(λ) or i(ν), except that the number of resonant modes is very large and the number of energy packets (photons) distributed over these modes is also very large in order to apply Stirling’s approximate formula, Equation (Equation 5). These conditions imply that the volume of the confined radiation is large with respect to its wavelength and that the energy intensity of the radiation is large with respect to the energy of a single photon, respectively. The Formulas (Equation 13) and (Equation 14) are, therefore, valid for any type of radiation i(λ) satisfying these conditions, not only for black-body radiation.

## 3. Derivation of Planck’s Radiation Law for the Spectrum of a Black-Body

### 3.1. Planck’s Radiation Law

For electromagnetic radiation in equilibrium with material at temperature *T* we can derive Planck’s radiation law for black-body emission using Equation (Equation 6) for the entropy,
(15)$$\frac{1}{T}=\frac{\partial S}{\partial E}=\frac{\partial S}{\partial U}\frac{\partial U}{\partial E}=\frac{1}{M}\frac{\partial S}{\partial U}=\frac{k}{\varepsilon }\ln\left(\frac{\varepsilon }{U}+1\right),$$
and solving for *U*,
(16)$$U\left(\lambda ,T\right)=\frac{hc}{\lambda }\frac{1}{\exp(hc/\lambda kT)-1},$$
which is the average energy emitted per resonant mode at wavelength λ. The total energy dE(λ,T) emitted by material at temperature *T* by all modes within a wavelength region between λ and λ+dλ is, therefore,
(17)$$dE\left(\lambda ,T\right)=\frac{dE(\lambda ,T)}{d\lambda }d\lambda =\frac{dE}{dM}\frac{dM}{d\lambda }d\lambda =U\left(\lambda ,T\right)\frac{dM}{d\lambda }d\lambda ,$$
where we have used E(λ,T)=M(λ)U(λ,T). Therefore, using Equations (Equation 16) and (Equation 3) in (Equation 17),
$$dE\left(\lambda ,T\right)=\frac{4n_{0}\pi V}{\lambda^{4}}\frac{hc}{\lambda }\frac{1}{\exp(hc/\lambda kT)-1}d\lambda =\frac{4n_{0}\pi hcV}{\lambda^{5}}\frac{1}{\exp(hc/\lambda kT)-1}d\lambda .$$

Finally, the energy density per unit volume per unit solid angle, per unit wavelength, is
(18)$$\rho \left(\lambda ,T\right)=\frac{1}{4\pi V}\frac{dE(\lambda ,T)}{d\lambda }=\frac{n_{0}hc}{\lambda^{5}}\frac{1}{\exp(hc/\lambda kT)-1},$$
and the energy flow per unit area of the body per unit second, per unit wavelength interval, is simply
(19)$$i\left(\lambda ,T\right)=c\rho \left(\lambda ,T\right)=\frac{n_{0}hc^{2}}{\lambda^{5}}\frac{1}{\exp(hc/\lambda kT)-1},$$
which is the Planck black-body radiation law.

Similarly, in terms of frequency ν,
(20)$$dE\left(\nu \right)=\frac{dE(\nu )}{d\nu }d\nu =\frac{dE}{dM}\frac{dM}{d\nu }d\nu =U\left(\nu \right)\frac{dM}{d\nu }d\nu ,$$
now,
$$\frac{dM}{d\nu }=\frac{dM}{d\lambda }\frac{d\lambda }{d\nu }=\frac{4n_{0}\pi V}{\lambda^{4}}\left(\frac{-c}{\nu^{2}}\right)=\frac{-4n_{0}\pi V\nu^{2}}{c^{3}}.$$

Therefore, this, in Equation (Equation 20), gives
(21)$$dE\left(\nu ,T\right)=\frac{-4n_{0}\pi V\nu^{2}}{c^{3}}\frac{h\nu }{\exp(h\nu /kT)-1}d\nu ,$$
and the energy density per unit volume per unit solid angle, per unit frequency, is
(22)$$\rho \left(\nu ,T\right)=\frac{1}{4\pi V}\frac{dE(\nu )}{d\nu }=\frac{n_{0}h\nu^{3}}{c^{3}}\frac{1}{\exp(h\nu /kT)-1},$$
and the energy flow per unit area of the body, per unit second, per unit frequency interval, is then
(23)$$i\left(\nu ,T\right)=c\rho \left(\nu ,T\right)=\frac{n_{0}h\nu^{3}}{c^{2}}\frac{1}{\exp(h\nu /kT)-1}.$$

Of course, the Planck black-body distributions, Equations (Equation 18), (Equation 19) and (Equation 23), assume that the material body is maintained at a uniform temperature *T* and that the body is transparent to the emitted radiation, which is only true for very low-density materials. For high-density materials, if the source of the energy is internal (such as with our Sun), the material may not be in thermodynamic equilibrium and the temperature *T* in Equations (Equation 18) and (Equation 22) must be taken as the surface temperature and the measured energy densities become per unit *surface area*, per unit solid angle, per unit wavelength (or frequency).

Note also that, for a wavelength-independent isotropy of the emitted or incident beams, the solid angle of the beam affects the entropy flux, or the energy flux, as a simple geometrical factor.

### 3.2. Energy Emitted by a Black Body at Temperature *T*

The total energy *E* emitted per unit volume per unit time into a 4π solid angle by a black-body at temperature *T* can be obtained by multiplying Equation (Equation 22) by 4π and integrating, over all frequencies ν,
(24)$$E=4\pi \cdot \frac{n_{0}h}{c^{3}}\int_{0}^{\infty }\frac{\nu^{3}}{\exp(h\nu /kT)-1}d\nu .$$

The integral $\int_{0}^{\infty }x^{3}/\left(\exp\left(x\right)-1\right)dx=3!\zeta \left(4\right)=\pi^{4}/15$, where ζ is the Riemann zeta function. Thus, Equation (Equation 24) becomes
(25)$$E=4\pi \frac{n_{0}\pi^{4}k^{4}T^{4}}{15c^{3}h^{3}}=\frac{4n_{0}\pi^{5}k^{4}}{15c^{3}h^{3}}T^{4}=\frac{n_{0}\pi^{2}k^{4}}{30c^{3}\hbar^{3}}T^{4}=\sigma T^{4},$$
where we have used ℏ=h/2π. Equation (Equation 25) is known as the Stefan–Boltzmann law and σ is the Stefan–Boltzmann constant.

The total entropy emitted by a black-body per unit volume, per unit time, into a 4π solid angle, is
(26)$$S=\frac{4}{3}\frac{E}{T}=\frac{4}{3}\frac{n_{0}\pi^{2}k^{4}}{30c^{3}\hbar^{3}}T^{3}=\frac{4}{3}\sigma T^{3}.$$

## 4. A Generalized Photon Force and Flow for the Photon Dissipation Process

In this section, from within the framework of the Classical Irreversible Thermodynamic theory developed by the school of Ilya Prigogine [1,2], we derive the entropy production for the dissipative irreversible process in which a wavelength region of the incident solar spectrum is absorbed and converted into heat by organic or inorganic material, as the product of a generalized force and flow, corresponding to, respectively, a wavelength dependent dissipative force F(λ) and a wavelength dependent rate (or flow) of photons N(λ) which become dissipated into the infrared spectrum.

The entropy produced per unit time due to the dissipative process of photon interaction with material contained within a well-defined boundary is simply the difference between the inwardly and outwardly directed flows of entropy, integrated over the boundary
(27)$$S=\int_{\Omega }\int_{A}\int_{\lambda }\left({\vec{j}}_{out}\left(\lambda \right)-{\vec{j}}_{in}\left(\lambda \right)\right)\cdot \hat{n}d\lambda dAd\Omega ,$$
where the integrals are over the wavelength λ dependence of the flows over the boundary surface *A*, with $\hat{n}$ being the unit normal to the surface and integrated over the solid angle Ω subtended by the radiation flow at the boundary. The entropy flows j(λ) have units, for example, of (WK${}^{-1}$ m ${}^{-3}$ s ${}^{-1}$ sr ${}^{-1}$) and *S* has units of (WK${}^{-1}$ s ${}^{-1}$).

Assuming that, after an initial relaxation, the system arrives at a stationary state (no change in time of the entropy of the system) and considering only closed systems (flow of energy over the boundary, but no flow of material), thereby ignoring all possibly coupled irreversible material processes acting over the boundary (such as, for example, convection, evaporation, etc.), the total entropy flow out of the boundary surface can be separated into a photon emitted $\vec{j^{e}}$, reflected $\vec{j^{r}}$ and transmitted $\vec{j^{t}}$, components
(28)$${\vec{j}}_{out}=\vec{j^{e}}+\vec{j^{r}}+\vec{j^{t}}.$$

For the incident entropy flow $j^{i}\left(\lambda \right)$ ( J K${}^{-1}$ m ${}^{-3}$ s ${}^{-1}$ sr ${}^{-1}$) per unit wavelength due to an arbitrary incident photon energy flow $i^{i}\left(\lambda \right)$, we use the expression obtained by Planck, derived in Section 2 (Equation (Equation 13)): (29)$$\begin{array}{ccc} j^{i}\left(\lambda \right) & = & \frac{n_{0}\;k\;c}{\lambda^{4}}\;\left[\;\left(1\;+\;\frac{\lambda^{5}\;i^{i}\left(\lambda \right)}{n_{0}\;h\;c^{2}}\right)\;\ln\;\left(1\;+\;\frac{\lambda^{5}\;i^{i}\left(\lambda \right)}{n_{0}\;h\;c^{2}}\right)-\left(\frac{\lambda^{5}\;i^{i}\left(\lambda \right)}{n_{0}\;h\;c^{2}}\right)\;\ln\;\left(\frac{\lambda^{5}\;i^{i}\left(\lambda \right)}{n_{0}\;h\;c^{2}}\right)\right], \end{array}$$
where $n_{0}$ denotes the polarization state, $n_{0}$ = 1 or 2 for polarized and unpolarized photons, respectively, *k* is the Boltzmann constant and *c* is the speed of light. For completeness, the corresponding expression in terms of frequency ν = c/λ, is (Equation (Equation 14)),
(30)$$\begin{array}{ccc} j^{i}\left(\nu \right) & = & \frac{n_{0}\;k\;\nu^{2}}{c^{2}}\;\left[\;\left(1\;+\;\frac{c^{2}\;i^{i}\left(\nu \right)}{n_{0}\;h\;\nu^{3}}\right)\;\ln\;\left(1\;+\;\frac{c^{2}\;i^{i}\left(\nu \right)}{n_{0}\;h\;\nu^{3}}\right)-\left(\frac{c^{2}\;i^{i}\left(\nu \right)}{n_{0}\;h\;\nu^{3}}\right)\;\ln\;\left(\frac{c^{2}\;i^{i}\left(\nu \right)}{n_{0}\;h\;\nu^{3}}\right)\right], \end{array}$$
per unit frequency interval, which has the units, for example, of (J K${}^{-1}$ m${}^{-2}$ sr${}^{-1}$).

The incident entropy flow per unit area at a given surface is, thus,
(31)$$J^{i}\;=\int_{\Omega }\int_{\lambda }j^{i}\left(\lambda \right)\cos\left(\theta \right)d\lambda d\Omega =\int_{\phi }\int_{\theta }\int_{\lambda }j^{i}\left(\lambda \right)\cos\left(\theta \right)\sin\left(\theta \right)d\lambda d\theta d\phi =g^{i}\left(\Omega \right)\int_{\lambda }j^{i}\left(\lambda \right)\;d\lambda ,$$
where θ is the angle of the normal of the surface to the incident beam and Ω is the solid angle subtended by the source at the surface, as shown in Figure 1. The geometrical factors, g(Ω), will be evaluated for different incoming and outgoing photon beams below.

Assuming that a fraction of the energy a(λ) is absorbed by the material within the boundary from the incident flow (the part that is not reflected or transmitted) is converted into an unpolarized black- or grey-body spectrum and emitted isotropically at an effective temperature $T_{e}$ (determined from energy conservation, see below), the emitted entropy flow out of the enclosing boundary for the photon dissipative process per unit wavelength dλ, per unit solid angle dΩ, per unit area dA, is
(32)$$\begin{array}{ccc} j^{e}\left(\lambda \right) & = & \frac{n_{0}\;k\;c}{\lambda^{4}}\;\left[\;\left(1\;+\;\frac{\lambda^{5}\;i^{e}\left(\lambda \right)}{n_{0}\;h\;c^{2}}\right)\;\ln\;\left(1\;+\;\frac{\lambda^{5}\;i^{e}\left(\lambda \right)}{n_{0}\;h\;c^{2}}\right)-\left(\frac{\lambda^{5}\;i^{e}\left(\lambda \right)}{n_{0}\;h\;c^{2}}\right)\;\ln\;\left(\frac{\lambda^{5}\;i^{e}\left(\lambda \right)}{n_{0}\;h\;c^{2}}\right)\right]. \end{array}$$

For the emitted energy flux $i^{e}\left(\lambda \right)$ used in $j^{e}\left(\lambda \right)$ of Equation (Equation 32), we assume a wavelength-dependent grey-body-emitted radiation at the effective temperature of $T_{e}$ determined by energy conservation (see below)
(33)$$i^{e}\left(\lambda ,T_{e}\right)=\epsilon_{m}\left(\lambda \right)\frac{n_{0}hc^{2}}{\lambda^{5}}\frac{1}{\exp(hc/\lambda kT_{e})-1},$$
where $\epsilon_{m}\left(\lambda \right)$ is the emissivity of the material as a function of wavelength λ.

Assuming that there is no transmitted radiation, or that both the reflected and transmitted radiations are Lambertian (i.e., scattered isotropically) and emitted into an equal solid angle, the reflected and transmitted part of the outgoing entropy flow can be summed together, $j^{r,t}=j^{r}+j^{t}$, and the entropy flow is
(34)$$j^{r,t}\left(\lambda \right)=j^{r,t}\left(i^{i}\left(\lambda \right)\left(1-a\left(\lambda \right)\right)\right),$$
and
(35)$$J^{r,t}\;=\;\int_{\Omega }\int_{\lambda }j^{r,t}\left(\lambda \right)\;d\lambda \cos\left(\theta \right)\;d\Omega =\;g^{r,t}\left(\Omega \right)\int_{\lambda }j^{r,t}(i^{i}\left(\lambda \right)\left(1-a\left(\lambda \right)\right)d\lambda ,$$
where we have assumed that the scattering isotropy is independent of the light wavelength.

Therefore, the total entropy production due to the absorption and dissipation process is
(36)$$\begin{array}{ccc} J & = & J^{e}+J^{r,t}-J^{i}\\ & = & \int_{\lambda }g^{e}\left(\Omega \right)\frac{n_{0}\;k\;c}{\lambda^{4}}\;\left[\left(1\;+\;\frac{\lambda^{5}\;i^{e}\left(\lambda \right)}{n_{0}\;h\;c^{2}}\right)\;\ln\;\left(1\;+\;\frac{\lambda^{5}\;i^{e}\left(\lambda \right)}{n_{0}\;h\;c^{2}}\right)-\left(\frac{\lambda^{5}\;i^{e}\left(\lambda \right)}{n_{0}\;h\;c^{2}}\right)\;\ln\;\left(\frac{\lambda^{5}\;i^{e}\left(\lambda \right)}{n_{0}\;h\;c^{2}}\right)\right]\\ & + & g^{r,t}\left(\Omega \right)\frac{n_{0}\;k\;c}{\lambda^{4}}\;\left[\left(1\;+\;\frac{\lambda^{5}\;i^{r,t}\left(\lambda \right)}{n_{0}\;h\;c^{2}}\right)\;\ln\;\left(1\;+\;\frac{\lambda^{5}\;i^{r,t}\left(\lambda \right)}{n_{0}\;h\;c^{2}}\right)-\left(\frac{\lambda^{5}\;i^{r,t}\left(\lambda \right)}{n_{0}\;h\;c^{2}}\right)\;\ln\;\left(\frac{\lambda^{5}\;i^{r,t}\left(\lambda \right)}{n_{0}\;h\;c^{2}}\right)\right]\\ & - & g^{i}\left(\Omega \right)\frac{n_{0}\;k\;c}{\lambda^{4}}\;\left[\left(1\;+\;\frac{\lambda^{5}\;i^{i}\left(\lambda \right)}{n_{0}\;h\;c^{2}}\right)\;\ln\;\left(1\;+\;\frac{\lambda^{5}\;i^{i}\left(\lambda \right)}{n_{0}\;h\;c^{2}}\right)-\left(\frac{\lambda^{5}\;i^{i}\left(\lambda \right)}{n_{0}\;h\;c^{2}}\right)\;\ln\;\left(\frac{\lambda^{5}\;i^{i}\left(\lambda \right)}{n_{0}\;h\;c^{2}}\right)\right]d\lambda . \end{array}$$
or, for completeness, in terms of frequency ν,
$$\begin{array}{ccc} J & = & J^{e}+J^{r,t}-J^{i}\\ & = & \int_{\nu }g^{e}\left(\Omega \right)\frac{n_{0}\;k\;\nu^{2}}{c^{2}}\left[\left(1\;+\;\frac{c^{2}\;i^{e}\left(\nu \right)}{n_{0}\;h\;\nu^{3}}\right)\;\ln\;\left(1\;+\;\frac{c^{2}\;i^{e}\left(\nu \right)}{n_{0}\;h\;\nu^{3}}\right)-\left(\frac{c^{2}\;i^{e}\left(\nu \right)}{n_{0}\;h\;\nu^{3}}\right)\;\ln\;\left(\frac{c^{2}\;i^{e}\left(\nu \right)}{n_{0}\;h\;\nu^{3}}\right)\right]\\ & + & g^{r,t}\left(\Omega \right)\frac{n_{0}\;k\;\nu^{2}}{c^{2}}\left[\left(1\;+\;\frac{c^{2}\;i^{r,t}\left(\nu \right)}{n_{0}\;h\;\nu^{3}}\right)\;\ln\;\left(1\;+\;\frac{c^{2}\;i^{r,t}\left(\nu \right)}{n_{0}\;h\;\nu^{3}}\right)-\left(\frac{c^{2}\;i^{r,t}\left(\nu \right)}{n_{0}\;h\;\nu^{3}}\right)\;\ln\;\left(\frac{c^{2}\;i^{r,t}\left(\nu \right)}{n_{0}\;h\;\nu^{3}}\right)\right]\\ & - & g^{i}\left(\Omega \right)\frac{n_{0}\;k\;\nu^{2}}{c^{2}}\left[\left(1\;+\;\frac{c^{2}\;i^{i}\left(\nu \right)}{n_{0}\;h\;\nu^{3}}\right)\;\ln\;\left(1\;+\;\frac{c^{2}\;i^{i}\left(\nu \right)}{n_{0}\;h\;\nu^{3}}\right)-\left(\frac{c^{2}\;i^{i}\left(\nu \right)}{n_{0}\;h\;\nu^{3}}\right)\;\ln\;\left(\frac{c^{2}\;i^{i}\left(\nu \right)}{n_{0}\;h\;\nu^{3}}\right)\right]d\nu . \end{array}$$

Simplifying Equation (Equation 36), by defining $b\left(\lambda \right)\equiv n_{0}hc^{2}/\lambda^{5}$, gives
(37)$$\begin{array}{ccc} J & = & \int_{\lambda }g^{e}\left(\Omega \right)i^{e}\left(\lambda \right)\frac{k\;\lambda }{hc}\left[\left(\frac{b(\lambda )}{i^{e}\left(\lambda \right)}+1\right)\;\ln\;\left(1\;+\;\frac{i^{e}\left(\lambda \right)}{b(\lambda )}\right)-\;\ln\;\left(\frac{i^{e}\left(\lambda \right)}{b(\lambda )}\right)\right]\\ & + & g^{r,t}\left(\Omega \right)i^{r,t}\left(\lambda \right)\frac{k\;\lambda }{hc}\left[\left(\frac{b(\lambda )}{i^{r,t}\left(\lambda \right)}+1\right)\;\ln\;\left(1\;+\;\frac{i^{r,t}\left(\lambda \right)}{b(\lambda )}\right)-\;\ln\;\left(\frac{i^{r,t}\left(\lambda \right)}{b(\lambda )}\right)\right]\\ & - & g^{i}\left(\Omega \right)i^{i}\left(\lambda \right)\frac{k\lambda }{hc}\left[\left(\frac{b(\lambda )}{i^{i}\left(\lambda \right)}+1\right)\;\ln\;\left(1+\frac{i^{i}\left(\lambda \right)}{b(\lambda )}\right)-\;\ln\;\left(\frac{i^{i}\left(\lambda \right)}{b(\lambda )}\right)\right]d\lambda . \end{array}$$

## 5. Entropy Production of an Arbitrary Photon Beam Interacting with Material

For the incident light $i^{i}\left(\lambda \right)$ coming from the Sun, we have the solid angle subtended by the slab at a distance $D_{ES}$ from the Sun (see Figure 2)
(38)$$\Omega \approx \frac{b^{2}}{4\pi D_{ES}^{2}}\cdot 4\pi =\frac{b^{2}}{D_{ES}^{2}},$$
where *b* is the dimension of the square slab.

The incident beam $i^{i}\left(\lambda \right)$ is assumed to be incident perpendicular to the slab (for example, the Sun, directly overhead and incident on a horizontal leaf). The geometrical factor for the incident light from the Sun is obtained by integrating over the disk of the Sun $\pi R_{S}^{2}$, where $R_{S}$ is the radius of the Sun, times the solid angle subtended by the slab at the Sun, divided by the surface area of the slab $b^{2}$ (since we are calculating the entropy production per unit surface area presented by the slab to the beam):(39)$$g^{i}\left(\Omega \right)=\pi R_{S}^{2}\cdot \frac{b^{2}}{D_{ES}^{2}}\cdot \frac{1}{b^{2}}=\pi \frac{R_{S}^{2}}{D_{ES}^{2}}.$$

For the incident energy flux $i^{i}\left(\lambda \right)$ used in the entropy flux $j^{i}\left(\lambda \right)$ of Equation (Equation 37), we assume grey-body radiation at the Sun’s surface temperature $T_{S}$ of 5779 K (see Equation (Equation 19))
(40)$$i^{i}\left(\lambda \right)=\epsilon_{S}\frac{n_{0}hc^{2}}{\lambda^{5}}\frac{1}{\exp(hc/\lambda kT_{S})-1},$$
where $\epsilon_{S}=0.987$ is the wavelength-independent emissivity of the Sun. To simulate the solar spectrum at Earth’s surface, after absorption in the atmosphere, we reduce the radius of the Sun until it reduces the solar constant from 1349 W m${}^{-2}$ to 1000 W m${}^{-2}$ at the surface (which happens at 87.54% of its actual radius if wavelength integration begins at 300 nm and ends at 1000 μm).

For the radiated beam from the slab of material (e.g., a leaf) acting as a grey-body emitting isotropically into both hemispheres (see Figure 2), the geometrical factor is
(41)$$g^{e}\left(\Omega \right)=2\cdot \int_{\Omega }d\Omega =2\cdot \int_{0}^{2\pi }\int_{0}^{\frac{\pi }{2}}sin\left(\theta \right)cos\left(\theta \right)d\theta d\phi =2\cdot \pi ,$$
where θ is the angle of the emitted beam (assumed Lambertian) to the normal of the surface of the slab.

For the emitted energy flux $i^{e}\left(\lambda \right)$ used in $j^{e}\left(\lambda \right)$ of Equation (Equation 37), we assume grey-body radiation at the slabs’s effective temperature of $T_{e}$
(42)$$i^{e}\left(\lambda \right)=\epsilon_{m}\left(\lambda \right)\frac{n_{0}hc^{2}}{\lambda^{5}}\frac{1}{\exp(hc/\lambda kT_{e})-1},$$
where $\epsilon_{m}\left(\lambda \right)$ is the emissivity of the material of the slab as a function of wavelength λ. We are approximating the material slab as a grey-body emitting into space at an effective temperature $T_{e}$ determined by energy conservation and the Stefan–Boltzmann Law (Equation (Equation 25))
(43)$$T_{e}={\left(\frac{E}{\epsilon_{avg}\sigma }\right)}^{\frac{1}{4}}={\left(\frac{g^{i}\left(\Omega \right)\int_{\lambda }a\left(\lambda \right)i^{i}\left(\lambda \right)d\lambda }{\epsilon_{avg}\sigma }\right)}^{\frac{1}{4}},$$
where σ is the Stefan–Boltzmann constant (Equation (Equation 25)) and where the total energy *E* is determined by integrating over the absorbed part of the incident spectrum.

The reflected $i^{r}\left(\lambda \right)$ and $i^{t}\left(\lambda \right)$ transmitted beams are assumed to be Lambertian, of the same wavelength dependence as the incident beam and emitted into one hemisphere, giving identical geometrical factors
(44)$$g^{r}\left(\Omega \right)=g^{t}\left(\Omega \right)=\int_{0}^{2\pi }\int_{0}^{\pi }sin\left(\theta \right)cos\left(\theta \right)d\theta d\phi =\pi .$$

Under these assumptions, the reflected and transmitted beams can be taken together, giving
(45)$$i^{r,t}\left(\lambda \right)=\left(1-a\left(\lambda \right)\right)\frac{R_{s}^{2}}{D_{ES}^{2}}i^{i}\left(\lambda \right),$$
and $g^{r,t}\left(\Omega \right)=\pi$.

## 6. Photon Force

In this section, we calculate the photon force at a given wavelength for the dissipative process as the entropy production divided by the wavelength-dependent photon flow. Such a force thus depends on the wavelength but not on the intensity of the incident light.

For the irreversible dissipative process, in which a part a(λ) of the incident photon beam is absorbed and converted into a black- or grey-body emitted spectrum, the first term in Equation (Equation 37), corresponding to the entropy flux of the emitted light beam, can be written, using Equation (Equation 26), as
(46)$$J^{e}=\frac{4}{3}\frac{\epsilon_{avg}}{T_{e}}\int_{\lambda }g^{i}\left(\Omega \right)i^{i}\left(\lambda \right)a\left(\lambda \right)d\lambda ,$$
where $T_{e}$ is the effective temperature determined using the energy absorption coefficient and energy conservation (Equation (Equation 43)), $\epsilon_{avg}$ is the average, energy-weighted, broadband emissivity of the material and a(λ) is the wavelength-dependent absorption coefficient.

The total entropy production per unit time per unit area, due to the interaction of the photon beam with the material slab, Equation (Equation 37), then becomes
(47)$$\begin{array}{ccc} J & = & \int_{\lambda }\frac{4\pi }{3}\frac{\epsilon_{avg}}{T_{e}}a\left(\lambda \right)i^{E}\left(\lambda \right)\\ & + & \pi \left(1-a\left(\lambda \right)\right)i^{E}\left(\lambda \right)\frac{k\;\lambda }{hc}\left[\left(\frac{b(\lambda )}{\left(1-a\left(\lambda \right)\right)i^{E}\left(\lambda \right)}+1\right)\;\ln\;\left(1\;+\;\frac{\left(1-a\left(\lambda \right)\right)i^{E}\left(\lambda \right)}{b(\lambda )}\right)-\;\ln\;\left(\frac{\left(1-a\left(\lambda \right)\right)i^{E}\left(\lambda \right)}{b(\lambda )}\right)\right]\\ & - & \pi i^{E}\left(\lambda \right)\frac{k\lambda }{hc}\left[\left(\frac{b(\lambda )}{i^{i}\left(\lambda \right)}+1\right)\;\ln\;\left(1+\frac{i^{i}\left(\lambda \right)}{b(\lambda )}\right)-\;\ln\;\left(\frac{i^{i}\left(\lambda \right)}{b(\lambda )}\right)\right]d\lambda , \end{array}$$
where $i^{E}\left(\lambda \right)={\left(R_{S}/D_{ES}\right)}^{2}\cdot i^{i}\left(\lambda \right)$ is the incident intensity of the solar spectrum at Earth’s surface, with $i^{i}\left(\lambda \right)$ obtained from the reduced radius of the Sun giving the solar constant at Earth’s surface of 1000 W m${}^{-2}$.

Using Equation (Equation 43) in Equation (Equation 47) and noting that the number of incident photons per unit area per unit time is simply
(48)$$N\left(\lambda \right)=i^{E}\left(\lambda \right)\cdot \lambda /hc,$$
we obtain
(49)$$J=\int_{\lambda }N\left(\lambda \right)\cdot F\left(\lambda \right)d\lambda ,$$
where the wavelength-dependent force for the dissipation process, F(λ), is
(50)$$\begin{array}{ccc} F(\lambda ) & \approx & \frac{hc}{\lambda }\frac{4\pi }{3}\frac{\epsilon_{avg}}{T_{e}}a\left(\lambda \right)\\ & + & \pi k\left(1-a\left(\lambda \right)\right)\left[\left(\frac{b(\lambda )}{\left(1-a\left(\lambda \right)\right)i^{E}\left(\lambda \right)}+1\right)\;\ln\;\left(1\;+\;\frac{\left(1-a\left(\lambda \right)\right)i^{E}\left(\lambda \right)}{b(\lambda )}\right)-\;\ln\;\left(\frac{\left(1-a\left(\lambda \right)\right)i^{E}\left(\lambda \right)}{b(\lambda )}\right)\right]\\ & - & \pi k\left[\left(\frac{b(\lambda )}{i^{i}\left(\lambda \right)}+1\right)\;\ln\;\left(1+\frac{i^{i}\left(\lambda \right)}{b(\lambda )}\right)-\;\ln\;\left(\frac{i^{i}\left(\lambda \right)}{b(\lambda )}\right)\right]. \end{array}$$

Note that the entropy production per unit time per unit area, Equation (Equation 49), has the usual Classical Irreversible Thermodynamic form of a sum over the flows times the forces. Notice also, from Equations (Equation 48) and (Equation 50), the non-linear relation between the force F(λ) of the dissipative process and the energy flow $i^{E}\left(\lambda \right)$ or the photon flow N(λ) from the incident to the outgoing spectra.

Equation (Equation 50) for the force F(λ) is plotted as a function of wavelength λ in Figure 3 (blue line). As expected, shorter photon wavelengths provide a greater force for photon dissipation (entropy production) than do longer wavelengths. This is analogous to the expectation that larger heat gradients lead to greater heat flow and entropy production than smaller gradients. Analogous also is the fact that, just as with greater temperature gradients, new physical processes can arise when physical barriers are surpassed (e.g., buoyant force overcoming frictional forces giving rise to convection) which promote the system into a new regime of greater heat flow and entropy production, in the case of photon interaction with material, shorter photon wavelengths can lead the molecular system to new covalent bondings (new pigments) which increase photon absorption and dissipation, something we have termed “microscopic dissipative structuring” [9].

The first term of Equation (Equation 50) is the contribution to the entropy production due to the absorption and emission (red line of Figure 3) and has a 1/λ dependence on wavelength.

The second term in Equation (Equation 50) is the contribution to the entropy production due to the reflected and transmitted part of the outgoing beam and is a constant (green line of Figure 3). This is because the reflected spectrum has the same wavelength dependence as the incident spectrum; the only factor that changes is the geometrical factor, which is independent of wavelength.

The last term of Equation (Equation 50) is the negative contribution of the entropy of the incident spectrum (grey line of Figure 3).

Also plotted in Figure 3 is the contribution to the entropy production, N(λ)·F(λ), for different wavelengths of the incident beam; the black line is for $i^{i}\left(\lambda \right)$ taken as a grey-body spectra at the Sun’s surface temperature, but normalized by reducing the Sun’s radius (see above) to give a solar constant of 1000 W m${}^{2}$ at Earth’s surface. Note that, for the grey-body solar spectrum, the maximum contribution to the entropy production corresponds to the wavelength bin at 502 nm and not at the peak photon intensity of the surface solar spectrum N(λ), which is at 635 nm, because, as the blue line of Figure 3 shows, the dissipation of shorter-wavelength photons has greater force F(λ) and, thus, leads to greater entropy production.

It is intriguing that the wavelengths of maximum absorption $\lambda_{max}$ of all important classes of plant and phytoplankton pigments (including chlorophyll) lie within wavelengths $430<\lambda_{max}<550$ nm (see Figure 4), since this is exactly where entropy production due to photon dissipation is maximal for our solar spectrum (see Figure 3), while photosynthetic efficiency is maximal between 600 and 700 nm (Figure 5), a fact established by targeted-spectrum LED light optimization for plant growth. The wavelength of the maximum number of incident photons at 635 nm corresponds, in fact, to the lesser peak of the red absorption of chlorophyll b. Energy absorbed between the wavelengths of 430 to 550 nm cannot be directly used in the reactive center of the photosynthetic system but must be dissipated into the red before being usable. These results, together, provide strong evidence for the proposition that the evolution of pigments, plants and ecosystems has been towards optimizing entropy production, not, as generally assumed, towards optimizing photosynthesis. It is, therefore, quite possible that not only chlorophyll, but most plant and phytoplankton pigments, were primordial dissipative pigments, even before the invention of photosynthesis. They were not, therefore, UV sunscreens for the photosynthetic system, and neither were they the result of UV survival selection [13], but rather they were self-organized dissipative structures which optimized their fundamental thermodynamic function of entropy production through photon dissipation.

**Figure 4 entropy-24-00076-f004:**
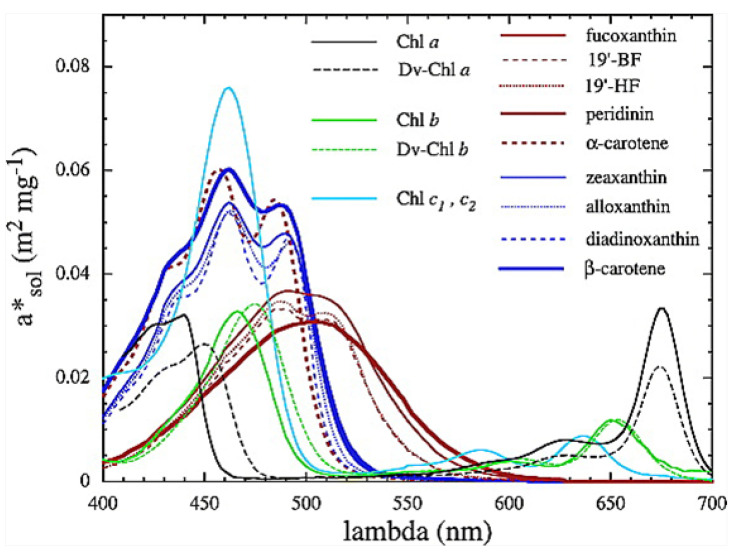
Phytoplankton pigment absorption spectra (m${}^{2}$ mg${}^{-1}$) from Bricaud et al. [14]. Absorption spectra of photosynthetic and nonphotosynthetic carotenoids are shown in red and blue, respectively. All pigments (including chlorophyll) have a wavelength of maximum absorption of $430<\lambda_{max}<550$ nm, exactly where entropy production due to photon dissipation is maximal for our solar spectrum (see Figure 5). Reprinted with permission from the American Geophysical Union.

## 7. Comparison of the Entropy Production of Inorganic and Dissipatively Structured Organic Material

Using Equation (Equation 43) in Equation (Equation 42) and then the equations for the light beams—(Equation 40), (Equation 42) and (Equation 45)—along with the respective geometrical factors together in Equation (Equation 37) allows the determination of the entropy production due to the interaction of an arbitrary incident light beam with a given slab of material, once the wavelength dependence of the absorption a(λ) and emissivity ϵ(λ) are given.

The wavelength dependence of a(λ) and ϵ(λ) for the different materials, both non-living and living, listed in Table 1, were estimated from experimental data published in the literature and are given in Figure 6, Figure 7, Figure 8, Figure 9, Figure 10 and Figure 11. Also given in the figures are the incident (blue) and the reflected plus the transmitted spectrum (violet), the difference being the part of the spectrum absorbed in the material. The red-shifted emitted light spectrum emanating from the material is also given (red).

The contributions to the entropy production for the incident $J^{i}$, absorbed $J^{e}$, reflected and/or transmitted $J^{r,t}$ beams, due to sunlight interacting with the different materials, as well as the total entropy production per unit area J, determined by using Equation (Equation 37), are given in Table 1.

The first column of Table 1 presents photon absorption and dissipation characteristics for a hypothetically perfect black-body (BB Ideal) material having maximum, and wavelength-independent, absorption of a(λ)=1.00 and maximum, and wavelength-independent, emissivity of ϵ(λ)=1.00. Such a perfect black-body gives the maximum possible entropy production, as can be verified from Equation (Equation 37), for any material interacting with sunlight. For a solar constant at Earth’s surface of 1000 W m${}^{-2}$ with the Sun overhead and perpendicular to the slab, the maximum entropy production of a black-body is 7.080 W m${}^{-2}$ K${}^{-1}$. The effective black-body temperature, obtained using Equation (Equation 43), under such a solar constant is 364.57 K (91.42 °C).

Column 2 gives the relevant data for a thick, 100 m layer of water. Most (98.8%) of the incident sunlight is absorbed in such a layer and since the long wavelength emissivity of water is also high (0.951), water of this thickness acts almost like a black-body and the entropy production due to the dissipation process (7.026 W m${}^{-2}$ K${}^{-1}$) is close to that of the perfect black-body (column 1).

For water of lesser thicknesses (columns 3 and 4), however, due to the transparency of water in the UV and visible spectra (∼300–700 nm, with peak transparency at 490 nm, see Figure 6, Figure 7 and Figure 8), much more of the light is transmitted and scattered (assumed to be Lambertian) rather than dissipated and the entropy production drops off to 4.898 W m${}^{-2}$ K${}^{-1}$ at 2 m thickness and to less than 1.545 W m${}^{-2}$ K${}^{-1}$ for a thickness corresponding to that of an average leaf (235 μm). This result for the thin 235 μm slab of water must be considered as an upper limit since, for such a thin layer, light scattering and transmission would be more specular rather than Lambertian, which results in less entropy production.

Column 5 of Table 1 gives the entropy production for sunlight interacting at noon with a desert of sand and small rocks. The wavelength-dependent absorption was determined from the albedo α(λ) as a(λ)=1−α(λ) using the experimental albedo data of Pinker and Karnieli et al. [17] for a semi-arid region of the Sahara. The long wavelength absorption was taken from the albedo data of Haapanala et al. [22]. The emissivity of a sand and pebble desert was determined from the data of Mattar et al. [18]. The entropy production of 4.714 W m${}^{-2}$ K${}^{-1}$ is an upper limit for a lifeless desert because even the most extreme deserts on Earth contain non-negligible amounts of organic pigments.

Column 6 gives the entropy production for an average leaf of thickness 235 μm (Figure 10). The data for absorption were taken from Gates [19]. A leaf has higher absorption, particularly at shorter wavelengths, than desert ground devoid of life (compare Figure 9 and Figure 10). Greater absorption at shorter wavelengths leads to greater entropy production for the leaf, even though the net amounts of energy absorbed by the leaf and desert are similar, as can be seen from Table 1. The photon force plotted in Figure 3 shows this more clearly. The entropy production is simply the photon force times the flow of photons. The higher emissivities of leaves also help to increase entropy production. Note that the actual contribution to the global entropy production attributed to the dissipative structure known as a leaf is actually about 8% higher than the value presented in Table 1, due to their contribution through transpiration to another coupled irreversible process, the water cycle [23].

Column 7 gives the entropy production of a climax conifer forest. The wavelength-dependent absorption was determined from the albedo data of Rautiainen et al. [21] for an 80+ year old strand of spruce trees and the emissivity data were obtained from Huang et al. [16]. The entropy production for such a forest, 6.834 W m${}^{-2}$ K${}^{-1}$, is close to the maximally obtainable entropy production for a perfect black-body and to what an ocean of depth of almost 100 m would achieve. This is an important indication of how the thermodynamic imperative of dissipative structuring guides evolution towards climax ecosystems (the conifer forest), including the water cycle and the animal life needed to support the dissipating trees, with near-perfect black-body characteristics and with near-maximum entropy production.

For comparison, the entropy production averaged over all of Earth’s surface is approximately 1.247 W m${}^{-2}$ K${}^{-1}$ and this occurs at an effective grey-body temperature of 273.0 K (−0.15 ${}^{\circ }$C) [24]. This much lower average value for the entropy production of the whole of Earth’s surface is due to the fact that the solar constant of 1000 W m${}^{-2}$ used in the above calculations is only valid at the Earth’s equator at noon. The solar constant, of course, decreases with latitude and goes to zero during the night. This reduces the average entropy production per unit area by a factor of 4. Further reducing the global average is the fact that a significant part of Earth is covered with material not as efficient at producing entropy as leaves or forests or deep water (e.g., clouds, the polar ice caps and dry grasses).

Contributing to the entropy production of Earth, in addition to the initial absorption of solar photons and their dissipation into heat in surface materials, are other secondary entropy-producing process occurring on Earth’s surface (e.g., water cycle, winds, ocean currents, hurricanes, etc.) which together contribute about 8% [25] to the whole Earth’s average value of 1.247 W m${}^{-2}$ K${}^{-1}$. Considering all positive and negative contributions to the global average entropy production, we find consistency between the values listed in Table 1 and the global average.

## 8. Discussion

Why natural systems, such as ecosystems, tend to evolve towards increasing their entropy production over time has not, so far, been discussed in this paper. For systems near to equilibrium, or even out of equilibrium but linear (with respect to generalized forces and flows), it can be rigorously shown that entropy production actually decreases over time if the external constraints are kept constant [1]. However, it is also known that far from equilibrium, kinetic factors (e.g., auto-catalysis) can compensate for the statistical improbability of self-organization and the system can evolve towards new, organized, stationary states of lower entropy but greater entropy production [2]. Furthermore, the evolution of real living systems depends strongly on their history or, in physics terms, their path through multidimensional phase space (making them non-Markovian), and they are strongly non-linear, resulting in a multitude of stationary states of different, possibly greater, entropy production which become available near bifurcations along a particular parameter of the system. In general, systems with positive feeback (auto-catalytic) tend to evolve toward greater entropy production. Under what physical conditions and exactly how these systems arise in general is the theme of ongoing analysis, but discussed at some length for microscopic dissipative structuring of molecular systems in our accompanying paper [5].

Analyses considering only the statistical aspects of out-of-equilibrium Markovian systems are blind to the dynamics (e.g., kinetic factors) and to history and, therefore, cannot adequately describe the repertoire of evolutionary responses available to non-equilibrium systems. In the Appendix of our accompanying article [5], we discuss in detail the relation between the statistical and thermodynamic approaches. The fact that a statistical analysis is inadequate to determine the evolution of such systems is, in fact, something that Prigogine and coworkers emphasized many years ago [2].

We have seen above that under a continuous light flux, the greatest entropy production is obtained by a material “self-organizing” into a material presenting characteristics as close to a black-body as possible (i.e., maximal absorption and emissivity). The naturally available abiotic material on Earth which approaches a black-body best under normally incident light is a thick layer of water (>100 m). However, for incident sunlight at angles less than perpendicular and for short UV and visible wavelengths, water becomes much more reflective than a leaf [26]. Therefore, integrating over an entire day and including the additional entropy production of vegetation due to their transpiration contribution to the water cycle [23], leaves and forests are greater entropy producers than pure water. To have similar entropy-producing potential as an average leaf of 235 μm thickness, for a normally incident photon beam, the layer of water would have to be at least 2 m thick (see Table 1).

The absorption of water at shallow incident angles and short wavelengths, however, can be increased if it contains organic pigments at its surface, such as phytoplankton (Figure 4), and this, most probably, was the thermodynamic imperative originating the dissipative structuring of organic pigments, and then life, on the ocean surface [7,9,23].

Sandy deserts and dry bare soil have a rather low emissivity which contributes to a low entropy production (Table 1), but emissivity increases from 1.7% to 16% when the water content becomes non-negligible, especially for sandy soils and in the 8.2–9.2-μm range where desert emission peaks (see Figure 9). It is known that forests produce low atmospheric pressure regions which bring water-laden air from the oceans further inland over continents than would normally be the case without forests [27], thus increasing further the entropy production over desert or barren land.

Due to fundamental physical limitations, all pigments have a finite physical size and a finite excited-state lifetime. Therefore, for any given leaf, only a finite number of photons can be absorbed per second per unit area. Given this physical limitation (which can, however, improve through dissipative structuring over evolutionary time) and given the wavelength dependence of the photon force for dissipation (Figure 3), a leaf absorbing short-wavelength photons would produce more entropy than one absorbing longer wavelengths. For our solar spectrum at Earth’s surface, pigments with maximum absorption within the range 430 to 550 nm will have maximal entropy production (black line of Figure 5) and this is, indeed, the case for most plant and phytoplankton pigments (Figure 4). If plants or phytoplankton were optimized for reproduction and growth (Darwinian theory), pigment absorption would be strongest between 600 and 700 nm where the incident number of photons is largest and where photosynthesis is most efficient. However, Figure 4 and Figure 10 clearly indicate that leaves absorb most strongly at short UV and visible wavelengths and, in fact, stop absorbing at the red-edge (∼700 nm) (presumably because of the limit to the photon handling rate due to the finite excited-state lifetime and physical size of the pigments); this is, therefore, an absorption which optimizes entropy production, not photosynthesis.

In support of this assertion, Wang et al. [28] have carried out experiments indicating that transpiration, rather than photosynthesis, is optimized in plants. Transpiration is directly related to photon dissipation since transpiration is driven by the resulting heat of photon dissipation at the leaf surface. As Wang et al. note in their Discussion, the liquid-to-vapor phase change is the most effective way to further dissipate the heat input into living systems. Our results presented above, and that of Wang et al., therefore, can be taken as evidence in favor of the *Thermodynamic Dissipation Theory of the Origin and Evolution of Life* [5,7,9], which suggests that life’s origin and evolution are the result of nature optimizing photon dissipation.

The wavelength dependence of the global entropy production of a planet may, therefore, be a useful and convenient biosignature for detecting life on planets of other stars. If the absorption of light in the atmosphere or surface of a planet is commensurate with the spectral region of maximal entropy production (short wavelength region) and as long as this region is not below approximately 200 nm, where successive ionization and destruction of carbon-based organic molecules occurs, then this would be good reason to suspect the occurrence of at least the first steps of life; the dissipative structuring of pigments on the planet. The total global entropy production of the planet, determined using the formalism presented here, may be an even simpler biosignature. For example, the global entropy production of Earth per unit incident energy and per unit area is significantly greater than that of its sister planets Venus and Mars [24] which are devoid of complex life. This is ostensibly due to the dissipatve structuring (evolution) of organic material (pigments in life) on Earth’s surface. Caution would have to be exercised, however, when employing such a biosignature if we are to distinguish between a surface of simple organic pigments (such as, for example, on Titan or Hadean Earth) and a rich organic biosphere containing a climax ecosystem with animals. Furthermore, as shown in Table 1, a planet covered entirely in deep pure water would have entropy-producing properties similar to that of complex ecosystems such as Earth’s. A sterile water world, however, would present other biosignatures, allowing its easy identification.

For physical conditions different from those on Earth (e.g., stellar spectra, pressure, temperature, pH, etc.), another type of “organic material” (not necessarily carbon-based) may be dissipatively structured under different wavelength regions, also giving them black-body-like properties. The possibilities could be very large and until this chemistry is adequately explored, the best bet we have for finding extraterrestrial life is looking for life similar to our own using entropy production as a biosignature and searching for planets similar to our own in size, density and at a similar distance from a similar G-type star. The James Webb telescope, in conjunction with other UV and visible telescopes, measuring wavelength-dependent albedos and emissivities, would be ideally suited to provide the data for these entropy-production calculations from the incident stellar UV/visible and emitted infrared planet spectra.

## 9. Conclusions

We have provided a modern derivation of the Planck’s formula for the entropy of an arbitrary beam of light. From this, we have determined the entropy production for the irreversible dissipative process in which part of the incident beam is absorbed in the material and re-emitted in a red-shifted grey-body spectrum, with the rest of the beam being either reflected or transmitted without change in the spectrum. Dividing the entropy production by the incident photon flow leads to an equation for the wavelength dependent force for photon dissipation or entropy production.

The greatest entropy production will be achieved through the dissipative structuring of material such that it obtains the characteristics of a black-body, i.e., maximal absorption and maximal emissivity. Such material could be either inorganic and naturally present in the environment (e.g., a thick layer of water), or dissipatively structured organic material under UV and visible light (e.g., organic pigments). We compared the entropy-producing potential of different inorganic and organic materials and found that leaves have greater entropy production than bare ground of sand and rock devoid of life and that climax forests have maximal entropy production, close to the black-body maximum. Sterile oceans of at least 100 m depth are good at entropy production but integrating over the entire day shows that forests and climax ecosystems and organic pigments floating on the ocean surface (e.g., phytoplankton) give even greater entropy production.

The greater dissipative potential of organic materials per unit volume, the fact that organic pigments in plants absorb short-wavelength photons in preference to longer wavelengths where photosynthesis is most efficient, and the fact that all major plant and phytoplankton pigments absorb very strongly wavelengths between 430 and 550 nm, where entropy production is optimal for our Sun, suggests that nature evolves towards increases in global entropy production and not towards optimizing photosynthesis or local reproduction efficacy. This provides strong evidence in favor of the *Thermodynamic Dissipation Theory for the Origin and Evolution of Life*.

The wavelength dependence of the global entropy production of an extraterrestrial planet could be a good indicator or biosignature of extraterrestrial life on its surface and should be relatively simple to implement with current and planned telescopes.

## Figures and Tables

**Figure 1 entropy-24-00076-f001:**
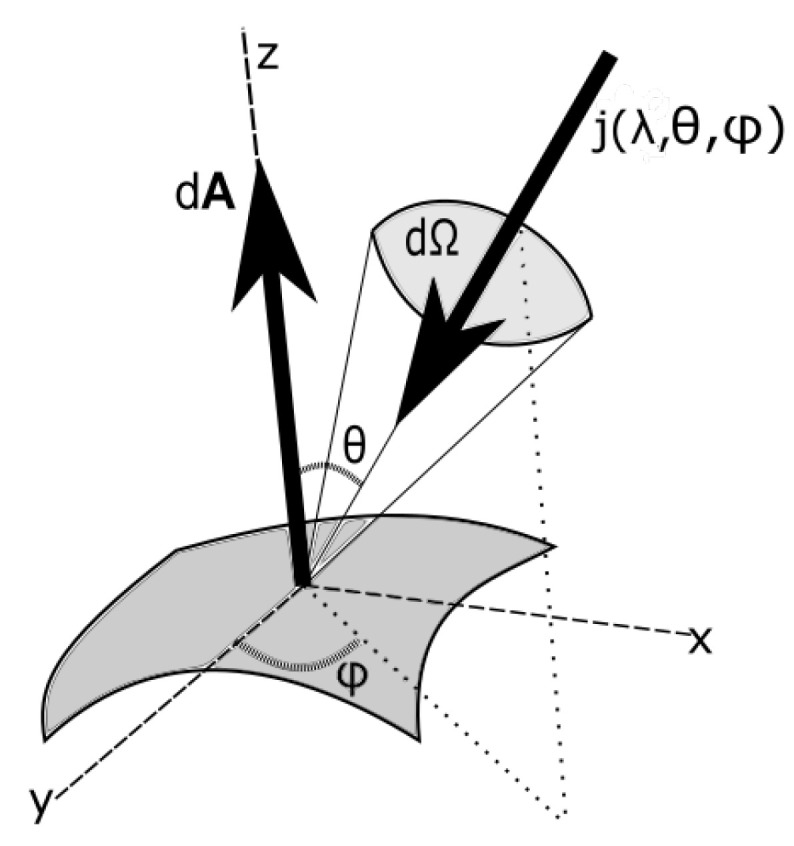
Diagram of incident radiation (energy i(λ,θ,ϕ) or entropy j(λ,θ,ϕ)) in solid angle dΩ over a surface element dA.

**Figure 2 entropy-24-00076-f002:**
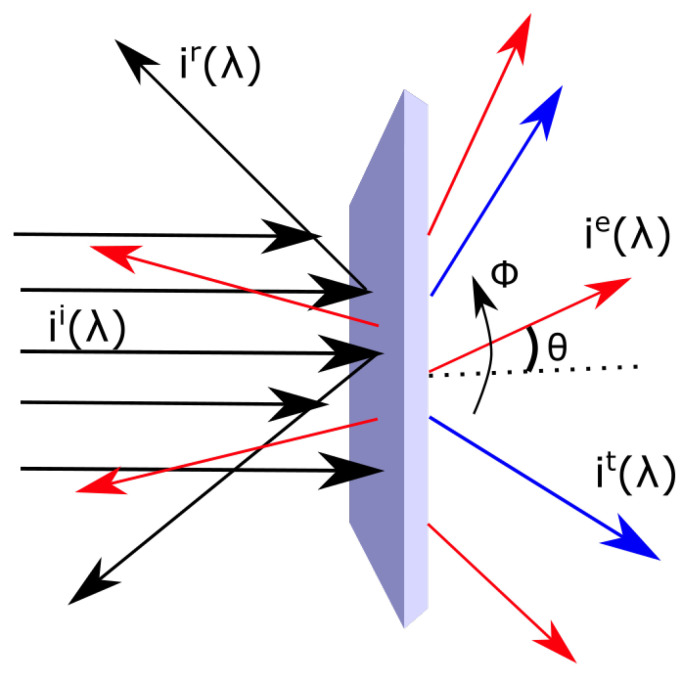
The light arriving in a parallel beam from a source $i^{i}\left(\lambda \right)$ and captured by the slab of material, with a fraction a(λ) absorbed and emitted isotropically as a black- or grey-body spectrum $i^{e}\left(\lambda \right)$ and a portion 1−a(λ) reflected or transmitted $i^{r,t}\left(\lambda \right)$ with the same wavelength dependence as the incident spectrum. θ is the angle of the incident, emitted, reflected or transmitted beams (the last two assumed to be Lambertian), with respect to the surface normal.

**Figure 3 entropy-24-00076-f003:**
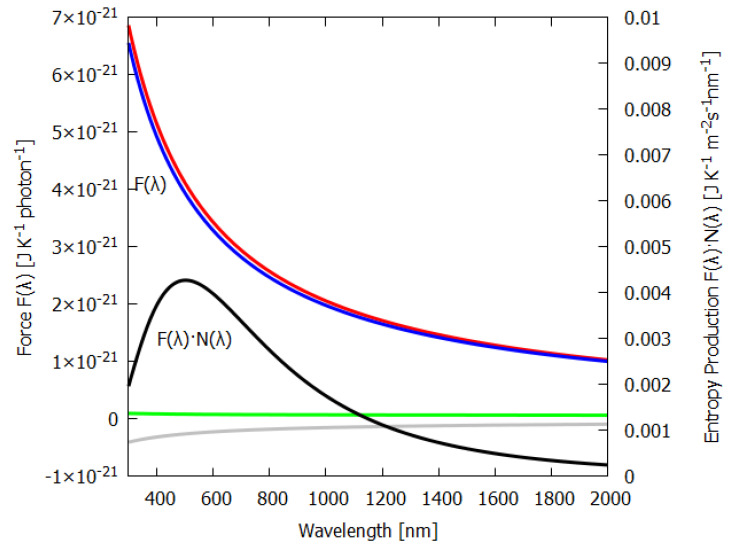
The thermodynamic force F(λ) (Equation (Equation 50)—blue line) as a function of wavelength λ for the irreversible process of photon dissipation of an incident beam arriving parallel to a slab of material (e.g., a leaf) with a fraction a(λ)=0.90 absorbed and emitted isotropically as a grey-body spectrum (emissivity $\epsilon_{avg}=0.980$) and a portion 1−a(λ)=0.10 reflected or transmitted with the same wavelength dependence as the incident spectrum. The contributions of the emitted (red line), reflected plus transmitted beams (green line) are assumed Lambertian. The grey line corresponds to the negative contribution of the incident light. Plotted on the right y-axis (black line) is the contribution to the entropy production, F(λ)·N(λ) (Equation (Equation 49)), per m${}^{2}$ per second integrated over a wavelength bin of 1 nm at the given wavelength for an incident energy spectrum $i^{E}\left(\lambda \right)$ of the Sun at Earth’s surface ( grey-body, absorption a(λ)=0.90, emissivity ϵ=0.98, solar constant 1000 Wm${}^{-2}$). Entropy production peaks at 502 nm for our solar spectrum. All major plant and phytoplankton pigments (chlorophyll a, b, carotenes, Xanthins, etc.) have their peak absorption between 430 and 550 nm (see Figure 4) and this is exactly where entropy production (black line) is maximized, whereas photosynthetic efficiency is maximal between 600 and 700 nm. This is a strong indication that plants have evolved to optimize photon dissipation rather than photosynthesis.

**Figure 5 entropy-24-00076-f005:**
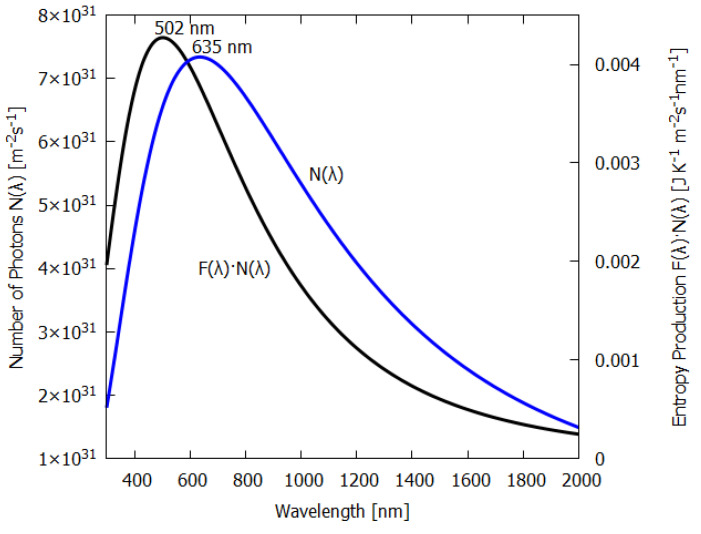
Plotted on the left y-axis (blue line) is the number of photons N(λ) incident at Earth’s surface per m${}^{2}$ per second for an incident energy spectrum $i^{E}\left(\lambda \right)$ of the Sun at Earth’s surface (grey-body, absorption a(λ)=0.90, emissivity ϵ=0.98, solar constant 1000 Wm${}^{-2}$). Plotted on the right y-axis (black line) is the contribution to the entropy production, F(λ)·N(λ), per m${}^{2}$ per second integrated over a wavelength bin of 1 nm at a given wavelength. Entropy production peaks at 502 nm, while photon intensity (number of photons) peaks at 635 nm for our solar spectrum.

**Figure 6 entropy-24-00076-f006:**
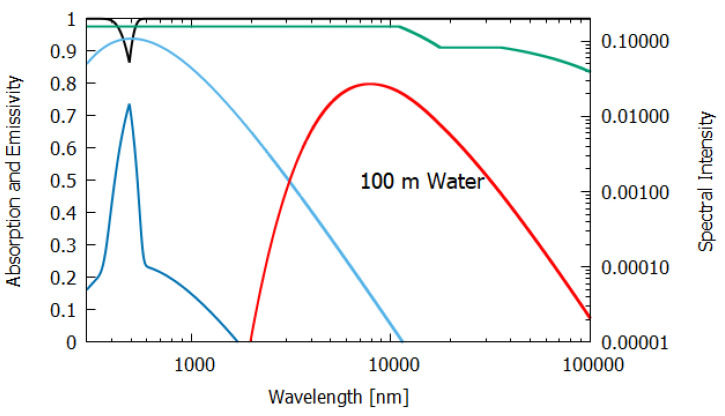
The absorption (black line) and emissivity (grey line) of water of 100 m thickness. Plotted also are the incident surface spectrum (blue) and the reflected and transmitted (violet) and emitted (red) spectra on a log–log scale (right y-axis). The data for the absorption are approximated from Chaplin [15] and the data for the emissivity are taken from Huang et al. [16].

**Figure 7 entropy-24-00076-f007:**
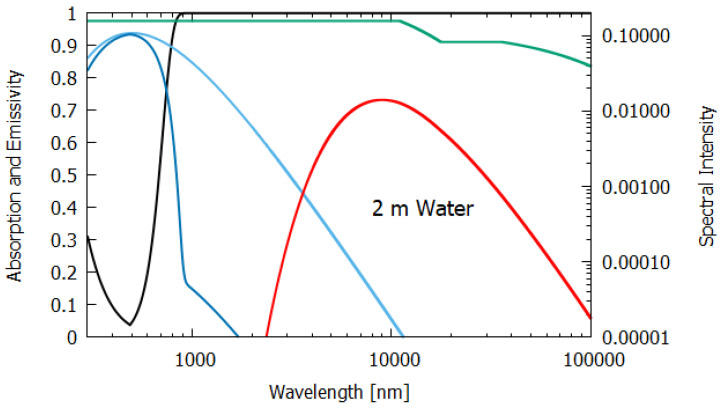
The absorption (black line) and emissivity (grey line) of water of 2 m thickness. Plotted also are the incident surface spectrum (blue) and the reflected and transmitted (violet) and emitted (red) spectra on a log–log scale (right y-axis). The data for the absorption are approximated from Chaplin [15] and the data for the emissivity are taken from Huang et al. [16].

**Figure 8 entropy-24-00076-f008:**
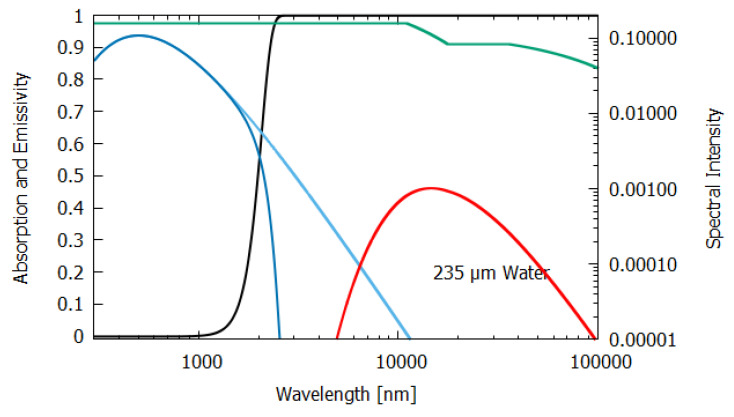
The absorption (black line) and emissivity (grey line) of water of 235 μm thickness. Plotted also are the incident surface spectrum (blue) and the reflected and transmitted (violet) and emitted (red) spectra on a log–log scale (right y-axis). The data for the absorption are taken from Chaplin [15] and the data for the emissivity are taken from Huang et al. [16].

**Figure 9 entropy-24-00076-f009:**
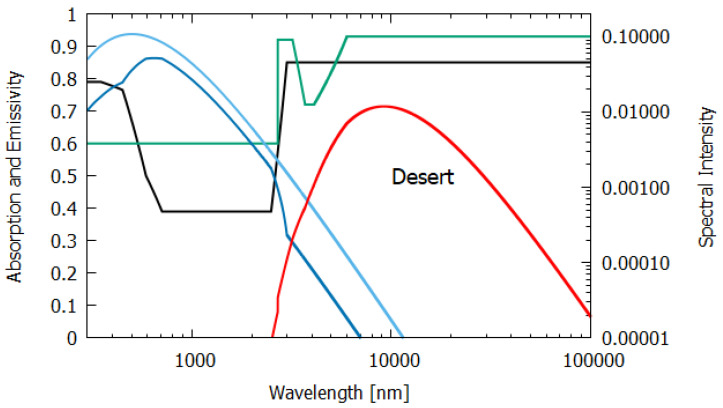
The absorption (black line) and emissivity (grey line) of a dry sand and pebble desert. Plotted also are the incident surface spectrum (blue) and the reflected and transmitted (violet) and emitted (red) spectra on a log–log scale (right y-axis). The data for the absorption are approximated from the experimental data of Pinker and Karnieli et al. [17] for a semi-arid region of the Sahara and the data for the emissivity are taken from Mattar et al. [18] for the Atacama desert.

**Figure 10 entropy-24-00076-f010:**
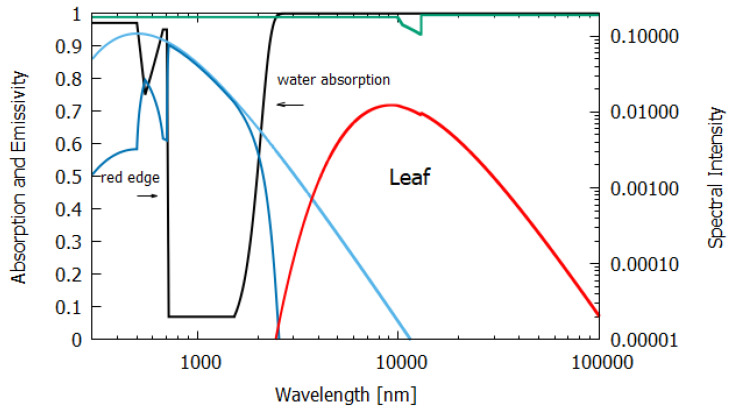
The absorption (black line) and emissivity (grey line) of a leaf. Plotted also are the incident surface spectrum (blue) and the reflected and transmitted (violet) and emitted (red) spectra on a log–log scale (right y-axis). The data for the absorption are approximated from Gates [19] and the data for the emissivity are taken from Ribeiro and Crowley [20] for a Cornus florida (Dogwood) leaf.

**Figure 11 entropy-24-00076-f011:**
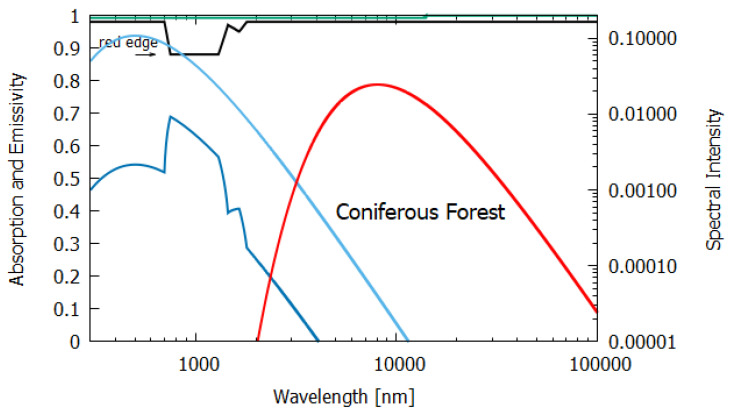
The absorption (black line) and emissivity (grey line) of a forest. Plotted also are the incident surface spectrum (blue) and the reflected and transmitted (violet) and emitted (red) spectra on a log–log scale (right y-axis). The data for the absorption are approximated from Rautiainen et al. [21] for an 80-year old coniferous spruce forest and the data for the emissivity are taken from Huang et al. [16].

**Table 1 entropy-24-00076-t001:** Entropy production due to the interaction of the solar spectrum at Earth’s surface with different materials. The incident solar spectrum is that of a grey-body at T =5779 K with emissivity 0.986 and adjusted for atmospheric absorption by reducing the solar radius (see text) to give a solar constant at Earth’s surface of 1000 W m${}^{-2}$. Effective temperatures $T_{eff}$ are obtained by using energy balance equations and the Stefan–Boltzmann law. Material temperature $T_{e}$ is obtained by including the average emissivity $\epsilon_{avg}$ (wavelength weighted by emitted photon energy). $J^{i}$ is the incident entropy flow, $J^{r,t}$ is the reflected and transmited entropy flow, $J^{e}$ is the red-shifted emitted entropy flow and J is the total entropy production per square meter of the material.

Material	Slab	H${}_{2}$O	H${}_{2}$O	H${}_{2}$O	Desert	Leaf	Forest
	BB Ideal	100 m	2 m	235 μm	Sand+Stone	235 μm	Conifer
$a_{avg}$	1.00	0.988	0.582	0.071	0.517	0.540	0.944
$\epsilon_{avg}$	1.00	0.951	0.945	0.919	0.926	0.984	0.994
$T_{eff}$ (K)	364.57	363.53	318.48	188.46	309.16	312.57	359.34
$T_{e}$ (K) (with $\epsilon_{avg}$)	364.57	368.10	322.99	192.50	315.15	313.79	359.83
Entropy Flux							
$J^{i}$ (W m${}^{-2}$ K${}^{-1}$)	0.233	0.233	0.233	0.233	0.233	0.233	0.233
$J^{r,t}$ (W m${}^{-2}$ K${}^{-1}$)	0.00	0.008	0.256	0.769	0.490	0.484	0.065
$J^{e}$ (W m${}^{-2}$ K${}^{-1}$)	7.312	7.250	4.875	1.009	4.456	4.608	7.002
J (W m${}^{-2}$ K${}^{-1}$)	**7.080**	**7.026**	**4.898**	**1.545**	**4.714**	**4.859**	**6.834**

## Data Availability

Not applicable.

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
