# Peer review of "A Photon Force and Flow for Dissipative Structuring: Application to Pigments, Plants and Ecosystems"

_entropy, 2022, doi:10.3390/e24010076_

Round 1

Reviewer 1 Report

The paper is very interesting and well written. I suggest some improvements:

  • I think that the authors must introduce one statement on the use of Boltzmann definition of entropy and not the use of the Gibbs definition of entropy
  • Beretta and Gyftopoulos have developed the analysis of entropy in electromagnetic fields: I suggest to quote them
  • Grisolia et al. has recently developed the consequences of irreversibility both in photon-electron interaction, time definition and in non-equilibrium temperature

After these improvements I suggest to accept the paper.

Author Response

We thank the reviewer for their kind remarks and their suggestions for improving the manuscript.

We have included the following statement concerning the Boltzmann and Gibbs definitions of entropy at line 87;

“Note that for non-interacting photons (true at least at normal photon densities and temperatures) the probabilities of all microstates are the same and the Boltzmann and Gibbs definitions of entropy are identical.”

We have included two new references to the analysis of the entropy of electromagnetic fields by Beretta and Gyftopoulos  and by Sieniutycz and Farkas  in the new version of the article at line 57;

“The thermodynamics of photon (electromagnetic field) interaction with material is still an active area of research \cite{EssexEtAl2005,BerettaGyftopoulos2015} which began with the work of Maxwell, Planck, and Einstein at the turn of the previous century.”

We have improved the introduction by including a new paragraph, starting on line 30, in which we cite a reference to justify the use of the local equilibrium hypothesis and also cite the interesting work of Grisolia et al.;

 “CIT theory assumes ``local equilibrium'' where the usual thermodynamic variables, as well as the Gibb´s relation between them, remain valid locally (in space and time) even though the global system may be far from equilibrium. Discussion of this assumption, in relation to the photon dissipative process, is given in an accompanying article in this special issue \cite{Michaelian2021}. Other relevant frameworks for treating non-equilibrium systems are discussed elsewhere \cite{LuciaGrisolia2021}.”

We have shortened the title and the abstract and made numerous improvements to the redaction by being more careful and concise. We have included 11 new references and removed 2 self-citations. We have included two new figures, Figs. 4 and 5, which aid in the explanation.

Reviewer 2 Report

I'm not a referee brought in to judge on the technical formulas.

The idea that life is set more at increasing entropy rather than at optimizing reproduction is not a particularly new one. See eg Prigogine or Gould who saw in the 2nd law of thermodynamics an arrow of time. Neither are referenced.

The use of "plants and ecosystems" reads as odd, what kind of life are we talking about here?

But I find the idea that increase in entropy can function as a biomarker intriguing. The idea however needs much more development. So major revisions.

Author Response

We thank the reviewer for their comments and suggestions on our manuscript. They have contributed significantly to improving our manuscript.

The theoretical framework used throughout our work is the Classical Irreversible Thermodynamic theory of Prigogine and coworkers. In the new version of the manuscript we have included 5 new references to the relevant works of Prigogine and coworkers [1-5]. 

We agree with the reviewer that the idea that life is about increasing entropy is not a new one. In fact, it goes back to Boltzmann in 1886. However, we are not presenting it as a new idea in the manuscript. Instead we set up the formalism to perform a careful calculation of the entropy production for photon interaction with materials, both abiotic and biotic, including the emitted, reflected and transmitted components of the outgoing light. We outline the conditions under which this entropy production is greater for biotic material than for abiotic material. This is important because it is a rather contentious issue with some claiming that photon dissipation (entropy production) is greater in abiotic material and not in biotic material [6]. The resolution of this is important to our hypothesis that living organisms optimize photon dissipation (entropy production) over photosynthesis or reproduction. We have carefully revised the redaction in the Abstract, and throughout the manuscript, to improve the presentation of the ideas we are expressing.

Both plants and ecosystems are photon absorbing dissipative systems. What we do in our manuscript, because we believe it relevant to our thesis, is to compare the entropy production of pigments in leaves, which receives and dissipates directly photons from the sun, to that of a climax forest ecosystem, which includes plants at different layers coupled to other dissipative processes like animals and the water cycle, which dissipate further still the incident photons.  In our new version, we have redacted more carefully the entire manuscript to make these comparisons more meaningful.

We further show in the manuscript that it is possible to derive a generalized force for the irreversible process of photon dissipation, which, as far as we are aware, has not been presented before in the literature. We argue that this force is strong enough, within a particular wavelength region, to be able to induce changes in molecular structure, without destroying the molecule through successive ionization, thereby leading to molecular configurations of greater dissipating efficacy. We show, in fact, that, due to the form of this force which derives from the incident solar spectrum at Earth’s surface, entropy production is maximum at around 502 nm, and this is very interesting because it is in this region where all important plant and phytoplankton pigments  absorbs strongest (see new Fig. 5). Besides new figures 4 and 5, we have included a new paragraph in the manuscript in the Section 6 Photon Force, line 233 and another in the Discussion section beginning on line 363 to emphasize these points.

These results, taken together, provide evidence that the evolution of natural organic materials (including ecosystems) has been towards increasing entropy production, and not, as generally assumed, towards optimizing photosynthesis or reproduction. We have now improved the discussion of this in the Discussion section, beginning at line 363. We also include a new reference to experimental evidence of the optimization of transpiration over photosynthesis in plants, Wang et al. This is in line with our proposition since transpiration is directly related to photon dissipation because transpiration is driven by the heat of photon dissipation in the leaves. We include in a new paragraph beginning on line 377 describing this.

Finally, we suggest that the determination of the entropy production of an extra-solar planet could be used as a biosignature for detecting life similar to our own on other planets. We have extended and improved our discussion of this in a paragraph beginning at line 385 in the Discussion section. The development of this idea, however, must be left to another article, but the main point is that it can be based on the formalism that we present in the manuscript.

We have shortened the title and the abstract and made multiple improvements to the redaction by being more careful and concise. We have included two new figures, Figs. 4 and 5, which aid in the explanation. We have removed 2 self-citations and included 11 new citations to others.

References:

  1. Prigogine, I. Introduction to Thermodynamics Of Irreversible Processes, third ed.; John Wiley & Sons, 1967.
  2. Prigogine, I.; Nicolis, G. Biological order, structure and instabilities. Quarterly Reviews of Biophysics 1971,

4, 107–144.

  1. Prigogine, I.; Nicolis, G.; Babloyantz, A. Thermodynamics of evolution. Physics Today 1972, 25, 23–28.
  2. Prigogine, I.; Nicolis, G.; Babloyantz, A. Thermodynamics of evolution. Physics Today 1972, 25, 38–44. doi:10.1063/1.3071140
  3. Glansdorff, P.; Prigogine, I. Thermodynamic Theory of Structure, Stability and Fluctuations;Wiley - Interscience, 505 1971.
  4. Björn, L. O.: Comment on K. Michaelian and A. Simeonov (2015) “Fundamental molecules of life are pigments which arose and co-evolved as a response to the thermodynamic imperative of dissipating the prevailing solar spectrum”, Biogeosciences Discuss., 2021-135, 2021. https://doi.org/10.5194/bg-

Reviewer 3 Report

I have serious reservations with this article by Michaelian and Mateo entitled: “A General Photon Force for Dissipative Structuring: Application to the Evolution of Plants and Ecosystems”. The authors utilize Prigogine’s irreversible thermodynamic theory “to question fundamental tenants of Darwinian theory and…provide support for the Thermodynamic Dissipation Theory for the Origin and Evolution of Life”. These are ambitious claims. In particular, they claim that their results “provide evidence that the evolution of plants and ecosystems has been towards optimizing entropy production, not, as generally assumed, towards optimizing photosynthesis”. These issues are profound and have been discussed in considerable detail over past decades though the authors ignore much of the relevant literature on the subject. In fact, the manuscript’s reference list of just 20 references ignores the leading contribution on the topic these past decades. Also, it is troubling that of the 20 references, almost half are self-citations.

The role of dissipative structures in life processes is now under intense study due to the discovery of chemical dissipative systems in 2010 (no references to this work appear). But one aspect of this debate is now beyond argument. It has now been categorically demonstrated that evolution does NOT optimize entropy production as the authors claim. See, for example, Ross et al., J. Phys. Chem. B 2012, 116, 7858−7865, so that argument has now effectively ended. Accordingly, I cannot recommend publication of the manuscript in its present form.

Author Response

We thank the reviewer for their useful comments and criticisms which have led to an improved version of our manuscript.

We agree with the reviewer that it was not appropriate to question the fundamental tenants of Darwinian Theory in our article. This occurred in two statements mentioning Darwin’s theory in the Introduction and the Conclusions. In the new version we have removed reference to this. Our paper, instead, concerns a detailed derivation of the entropy production through photon dissipation in material, showing that it is greater in biotic material than in abiotic material, a fact which has been somewhat contentious [1]. We are thus led to a result suggesting that biology tends to optimize photon dissipation at Earth’s surface.  In the new version of the manuscript, we have included a reference to an experimental paper [2] arguing that, rather than photosynthesis, transpiration is optimized in plants. This is related to photon dissipation since it is the heat of photon dissipation in the surface of the leaf that drives transpiration. This reference has been included in a new paragraph beginning at line 377 in the Discussion section of the new version of the manuscript.

Since the reviewer only mentioned a date, “2010”, without giving a reference concerning chemical dissipative systems, we are not sure which work he/she is referring to. We can say, however, that chemical dissipative systems have been suggested to be important in life and evolution beginning with  Prigogone and coworkers in the 1950’s [3-6]. These references are now included in the Introduction of the revised version of the manuscript (line 27). We emphasize, however, that our system is not a chemical dissipative system, but rather a photon dissipative system.

By comparing the photon dissipation results for biotic and abiotic systems, we suggest in the manuscript that ecosystems tend to increase the dissipation of photons on Earth and that this is related to increases in the entropy production of the biosphere. We do not rely on, nor employ, the maximum entropy production principle MEPP (criticized by the Ross et al. article cited by the reviewer), nor do we comment on its validity.

We share the reviewer’s and Ross et al.´s skepticism concerning the MEPP since Prigogine showed long ago that a non-equilibrium system can evolve to either greater or lesser entropy production [3]. There is no general potential that can be optimized to determine the evolution of an arbitrary system out of equilibrium. For systems near to equilibrium, or even out of equilibrium but linear (with respect to generalized forces and flows), it can be rigorously shown that entropy production actually decreases over time if the external constraints are held constant [3]. However, it is also known that far from equilibrium, kinetic factors (e.g. autocatalysis) can compensate for the statistical improbability of system self-organization and thus the system evolves towards new stationary states of lower entropy and, interestingly, greater entropy production [7]. ). In these situations, with multiple stationary states available to the system with different entropy production, the system will evolve through fluctuations near a bifurcation to other stationary states, but those stationary states with greater stability tend to be those states with greater entropy production.

This is the case for autocatalytic chemical systems as was shown by Glansdorff and Prigogone [7]. Under what physical conditions, and exactly how, these systems arise in general is the theme of ongoing analysis and is discussed at some length for photochemical microscopic dissipative structuring of molecular systems in our accompanying paper [8]. We now make reference to this in a new paragraph of the Introduction starting at line 30 and another in a paragraph of the Discussion starting at line 322.

Furthermore, the evolution of real living systems depends strongly on their history or, in physical terms, their path through multidimensional phase space, making them non-Markovian.  Analyses considering only the statistical aspects of out of equilibrium Markovian systems, are blind to the dynamics (e.g. kinetic factors) and history, and therefore cannot correctly describe the full repertoire of evolutionary responses available to these systems. In the Appendix to our accompanying article [8] we discuss in detail the relation between the statistical and thermodynamic approaches. The fact that a statistical analysis is insufficient to determine the evolution of such systems is, in fact, something that Prigogine and coworkers emphasized many years ago [7]. This is briefly summarized in the Discussion section of the new version of our manuscript beginning on line 322.

We have included 11 new references in the new version addressing the points of the reviewer.

We have shortened the Title and the Abstract and made multiple improvements to the redaction by being more careful and concise. We have included two new figures, Figs. 4 and 5, which aid in the explanation.

References

[1] Björn, L. O.: Comment on K. Michaelian and A. Simeonov (2015) “Fundamental molecules of life are pigments which arose and co-evolved as a response to the thermodynamic imperative of dissipating the prevailing solar spectrum”, Biogeosciences Discuss. [preprint], https://doi.org/10.5194/bg-2021-135, in review, 2021.

[2] Wang, J., Bras, R. L., Lerdau, M., and Salvucci, G. D. (2007), A maximum hypothesis of transpiration, J. Geophys. Res., 112, G03010, doi:10.1029/2006JG000255.

[3] Prigogine, I. Introduction to Thermodynamics Of Irreversible Processes, John Wiley & Sons, third edition, 1967.

[4] I. Prigogine and G. Nicolis (1971), Biological order, structure and instabilities, Quarterly Reviews of Biophysics, 4, 107-144.

[5] I. Prigogine and G. Nicolis and A. Babloyantz, Thermodynamics of evolution, Phys. Today, 1972}, 25, 11, 23-28, https://doi.org/10.1063/1.3071090

[6] I. Prigogine and G. Nicolis and A. Babloyantz, Thermodynamics of evolution, Phys. Today, 1972}, 25, 12, 38-44, https://doi.org/10.1063/1.3071140

 [7] Glansdorff, P. and Prigogine, I., Thermodynamic Theory of Structure, Stability and Fluctuations, Wiley – Interscience, 1971.

[8] Michaelian, K. The Dissipative Photochemical Origin of Life: UVC Abiogenesis of Adenine.  Entropy 202123, 217. https://doi.org/10.3390/e23020217

Round 2

Reviewer 1 Report

I think that the paper has been improved.

Reviewer 3 Report

Following the authors' revisions and explanations as well as a more comprehensive reference list, I can now support publication of the revised version.